# A Novel Method for Pose and Position Calibration of Laser Displacement Sensors

**DOI:** 10.3390/s23041762

**Published:** 2023-02-04

**Authors:** Liya Han, Long Yu, Xusheng Zhu

**Affiliations:** AVIC ChengDu Aircraft Industrial (Group) Co., Ltd., Chengdu 610092, China

**Keywords:** laser displacement sensor, surface normal measurement, pose and position calibration

## Abstract

Laser displacement sensors are widely used in the aviation industry for the purpose of surface normal measurements. The measurement of a surface normal depends on prior knowledge of the poses and positions of the sensors, which are obtained through calibration. This paper introduces a new parameter to the traditional calibration procedure, to reduce the calibration error, and explores the factors affecting calibration using the Monte Carlo method. In the experiment, the normal measurement error of the probe consisted of four sensors after calibration was less than $0.1^{\circ }$, which satisfied the established requirements. This paper indicates the boundary conditions for a successful calibration and validates the proposed method, which provides a new method for the pose and position calibration of laser displacement sensors and other similar sensors.

## 1. Introduction

In the aircraft manufacturing industry, drilling holes in freeform surfaces is a widespread requirement. The verticality of the hole, respective to the surface around it, is related to the strength of the connection and its safety to ensure long-term service. This derives the need for surface normal measurements, which are usually achieved by using laser displacement sensors, for their agility, accuracy, and ease of integration [1]. The laser displacement sensor (LDS) is an optical sensor that works under the triangulation measurement principle and returns the displacement value of the measured point relative to the origin [2]. The measurement is achieved through visual identification of the projection point of a laser beam on the surface of the measured object (i.e., the measured point) and the origin refers to the point at which the laser beam is emitted. As a sensor for one-dimensional measurements, LDSs are widely used in situations in which only the relative linear distance is needed [3,4,5]. In these scenarios, the origin of the sensor need not be known. However, in some other situations [6,7,8], LDSs work as a probe attached to a measurement system and the 3D position of the laser projection point needs to be defined, which requires prior knowledge of the direction and origin of the laser beam in an external coordinate system (usually the measurement system). Methods have been developed and can be divided into two main categories: the so-called sphere calibration [9,10,11] and the so-called plane calibration [12,13]. The main idea of the above methods is to establish an expression of the laser projection point on a specific target, such as by substituting the point into the equation of a sphere or plane.

In practice, the sphere calibration method is mainly used for single-sensor calibration, whereas the plane calibration method is mainly used for multi-sensor calibration. In the former case, the sensor is mounted to coordinate systems, such as measuring machines or machine tools, and the calibration is based on the original coordinates of the coordinate systems. The sensor approaches the sphere from different directions and obtains distance values, and the different measurements are constrained by a common static sphere. The raw coordinates of the coordinate system are the basis for building the equations. In the latter case, the sensors are formed into a diamond array and mounted to a robot, or machine-tool-drilling system, for surface normal measurements. When the 3D coordinates of the four measured points of the LDS are obtained, the surface normal can be calculated via vector fork multiplication, plane fitting, or curved surface fitting. Thus, the sensors are expected to be calibrated under a unified coordinate system. A sketch of the surface normal measurement scene is shown in Figure 1. As shown, four LDSs labeled with $LDS_{1}$ to $LDS_{4}$ are combined. $O_{1}$ to $O_{4}$ are the origin points of the LDSs, and $P_{1}$ to $P_{4}$ are the laser projection points. Only if the coordinates of $P_{1}$ to $P_{4}$ are obtained under a unified coordinate system O-XYZ, can the surface normal of the surface to be measured be calculated. The calculation method depends on whether the area being measured should be treated as a plane or a surface with curvature. Therefore, our goal was to calibrate the three-dimensional coordinates of $O_{1}$ to $O_{4}$ in the O-XYZ coordinate system, as well as the orientation vector of each laser beam.

In the plane calibration method, the sensor measures the same flat plate at different attitudes, and the attitude of the flat plate is simultaneously acquired by using other measuring devices. The most used measuring device is a laser tracker. Yu [12] calibrated the LDSs to work as a surface normal sensor of a machine tool drilling system with the traditional plane calibration. In this work, planar equations were established using the unknown laser beam vector and laser beam origin. Then, the system of equations was solved with the least squares method. The maximum angular deviation in drilling was $0.44^{\circ }$ under the guidance of the calibrated sensor. The method was simple and clear, but also sensitive to measurement errors in calibration. In a similar scenario, Chen [13] constructed a nonlinear target expression, based on a planar equation and used an extended Kalman filter for the solution. It had a maximum angular deviation of $0.1780^{\circ }$ after calibration and reduced the angle error in drilling, but required a relatively precise estimate of the initial values of LDS parameters. Furthermore, Kuester [14] calibrated a series of LDSs around a blade in the wind tunnel by using prior knowledge of the blade shape and its rotation angle; the uncertainty in the rotation measurement was $0.032^{\circ }$, but the scenario was limited to the attack angle measurement of the model in the wind tunnel.

For surface normal measurements, there are three reasons to use plane calibration instead of sphere calibration:The surface normal needs to be additionally measured, which usually means that the raw positioning of the platform is not capable of handling a large working space range, limiting the utilization of original data of the coordinate systems;The simultaneous calibration of four sensors requires a large standard sphere (in order to be “seen” by four sensors at the same time), which is difficult to manufacture, whereas plates with a datum plane of the corresponding size are much less difficult to manufacture;The relative position and attitude of the sensor and target need to be changed and measured during calibration, and the plane is easier to measure than a sphere when the original coordinate data cannot be relied upon.

The plane calibration method establishes equations of the datum plane both from the sensors and other measuring devices, such as laser trackers. Each plane position and attitude generates an equation, of which a certain amount is solvable for the laser beam direction vector and origin coordinates of each laser displacement sensor.

Although the plane calibration method is simple and easy to implement, it still has some problems:Since both the sensors and the other measuring devices observe the same side of the plate, an additional area is required to avoid occlusion, which increases the difficulty of manufacturing and guaranteeing the accuracy of the plate;Since some parameters in the calibration equation are coupled (this is shown in Section 2), the calibration process is more sensitive to measurement errors and requires more equations to suppress;Since the laser beam direction vector and sensor origin coordinates are difficult to physically contact (they just exist algorithmically), it is difficult to evaluate the calibration results objectively, such as through verification by a three-coordinate machine.

In an attempt to calibrate a probe that measures the surface normal (the probe is described in detail in Section 4), we found that the existing method did not allow us to achieve a measurement error of less than $0.1^{\circ }$. The measurement error here is defined as the angle between the measured surface normal and the actual surface normal, the latter of which is obtained by a structured light [15] probe.

In this paper, a novel method for the pose and position calibration of laser displacement sensors is proposed, with the aim of solving the three problems. This method realizes the calibration of the laser beam direction vector and sensor origin coordinates based on a double-side-grinded monocrystalline silicon wafer, and evaluates the calibration results with high accuracy. The basic idea is that a double-side-grinded monocrystalline silicon wafer is placed between the LDSs and a structured light probe (SLP). When the pose of the wafer is changed, the SLP measures its pose and position by scanning the wafer, and then a group of linear equations is constructed. Each time the wafer is moved, a group of equations is established, and, once there are enough equations, they can be solved by using the least squares method. For systems of super-determined linear equations with errors, the least squares method is a major method of flattening the error, and a sufficient number of equations can suppress the error to an ideal degree [16]. The exact number of equations for which the desired result is obtained are later indicated in subsequent simulations. The thickness of the wafer is measured and used to remove unsuitable equations. The remaining equations are used to solve the calibration parameters. The main merits include the following:This method simplifies the calibration scene. For manual scenarios, only sensors, wafers, and probes capable of measuring planes are required, and there are no requirements for the movement of the wafers. Additionally, the size of the wafer is limited to only cover the range of sensors to be calibrated, which further reduces the cost.The computational overhead is small, requiring only a linear solution to achieve sufficient accuracy, and, thus, it can be easily integrated into embedded systems.The error of calibration can be indicated by the difference between the wafer thickness obtained by calibration and the actual high-accuracy thickness measurement, which is easy to derive.

The remainder of this paper is organized as follows: In Section 2, the calibration method is modeled. In Section 3, the calibration method is examined through simulations to explore the boundary conditions for a successful calibration and to validate its superiority to the traditional plane method. In Section 4, the calibration and evaluation experiment is carried out. The conclusion is given in Section 5.

## 2. LDS Calibration Model

In the existing plane calibration methods, representation of the datum plane is the key to establishing an equation. If the sensor origin coordinates and laser beam direction vector are assumed to be ($x_{0}$, $y_{0}$, $z_{0}$) and (l,m,n), and the voltage value proportional to the distance is *v*, then the laser projection point can be represented as ($x_{0}$ + *lv*, $y_{0}$ + *mv*, $z_{0}$ + *nv*). It should be pointed out that when the return value of the LDS is measured in voltages, the direction vector here coupled the conversion coefficient of voltage to physical distance. Therefore, it would no longer be a unit vector and the length of the vector (l,m,n) is the voltage–physical distance conversion coefficient. When the laser projection point falls on a plane, it should satisfy the plane equation obtained by other measuring devices:(1)$$A\left(x_{0}+lv\right)+B\left(y_{0}+mv\right)+C\left(z_{0}+nv\right)+D=0$$

In which *A*, *B*, *C* and *D* are the parameters of planar equation measured by other devices. It can be seen here that the six coefficients A,Av,B,Bv,C, and *Cv* in the equation were obtained by coupling four known quantities A,B,C, and *v*, which meant that the measurement error of A,B,C, and *v* would have a greater impact on the solution of the system of equations than if the coefficients were independent. In our method, the additionally measured plane and the laser-projected plane were not the same. They were planes of opposite sides of the wafer. Thus, if we assumed that the thickness of the wafer was *d*, Equation (Equation 1) could be rewritten as:(2)$$A\left(x_{0}+lv\right)+B\left(y_{0}+mv\right)+C\left(z_{0}+nv\right)+\sqrt{A^{2}+B^{2}+C^{2}}d=-D$$

Here, *d* was treated as an unknown variable rather than a known constant, which made subsequent optimization and validation possible.

After changing the wafer position and attitude several times, sufficient equations were established and the system of linear equations was obtained, in which $A_{r}$, $B_{r}$, $C_{r}$, and $D_{r}$ were the measured plane parameters in operation *r*, and $v_{r}$ was the voltage read from any of the sensors being calibrated, as follows:(3)$$\left[\begin{array}{ccccccc} A_{1} & A_{1}v_{1} & B_{1} & B_{1}v_{1} & C_{1} & C_{1}v_{1} & \sqrt{A_{1}^{2}+B_{1}^{2}+C_{1}^{2}}\\ A_{2} & A_{2}v_{2} & B_{2} & B_{2}v_{2} & C_{2} & C_{2}v_{2} & \sqrt{A_{2}^{2}+B_{2}^{2}+C_{2}^{2}}\\ \cdots & \cdots & \cdots & \cdots & \cdots & \cdots & \cdots \\ A_{r} & A_{r}v_{r} & B_{r} & B_{r}v_{r} & C_{r} & C_{r}v_{r} & \sqrt{A_{r}^{2}+B_{r}^{2}+C_{r}^{2}}\\ \cdots & \cdots & \cdots & \cdots & \cdots & \cdots & \cdots \\ A_{s-1} & A_{s-1}v_{s-1} & B_{s-1} & B_{s-1}v_{s-1} & C_{s-1} & C_{s-1}v_{s-1} & \sqrt{A_{s-1}^{2}+B_{s-1}^{2}+C_{s-1}^{2}}\\ A_{s} & A_{s}v_{s} & B_{s} & B_{s}v_{s} & C_{s} & C_{s}v_{s} & \sqrt{A_{s}^{2}+B_{s}^{2}+C_{s}^{2}} \end{array}\right]•\left[\begin{array}{c} x_{0}\\ l\\ y_{0}\\ m\\ z_{0}\\ n\\ d \end{array}\right]=\left[\begin{array}{c} -D_{1}\\ -D_{2}\\ \cdots \\ -D_{r}\\ \cdots \\ -D_{s-1}\\ -D_{s} \end{array}\right]$$

The calibration calculations are independent for each sensor, and thus, each wafer position and attitude can be used by multiple sensors simultaneously. The minimum value of *s* was 7 to solve equation system (3). However, for more accurate results, *s* should be as large as possible and solved using the linear over-determined system of the equations’ solution. Note that if $A_{s}$, $B_{s}$, and $C_{s}$ were unitized (i.e., $\sqrt{A_{s}^{2}+B_{s}^{2}+C_{s}^{2}}=1$) in the method (e.g., if you obtained *A, B, C,* and *D* by using a monocular camera and a checkerboard pattern), then they should be multiplied by a random number to guarantee that the equation would be solvable. If we represent the coefficient matrix in terms of *Q*, i.e.: (4)$$Q=\left[\begin{array}{ccccccc} A_{1} & A_{1}v_{1} & B_{1} & B_{1}v_{1} & C_{1} & C_{1}v_{1} & \sqrt{A_{1}^{2}+B_{1}^{2}+C_{1}^{2}}\\ A_{2} & A_{2}v_{2} & B_{2} & B_{2}v_{2} & C_{2} & C_{2}v_{2} & \sqrt{A_{2}^{2}+B_{2}^{2}+C_{2}^{2}}\\ \cdots & \cdots & \cdots & \cdots & \cdots & \cdots & \cdots \\ A_{r} & A_{r}v_{r} & B_{r} & B_{r}v_{r} & C_{r} & C_{r}v_{r} & \sqrt{A_{r}^{2}+B_{r}^{2}+C_{r}^{2}}\\ \cdots & \cdots & \cdots & \cdots & \cdots & \cdots & \cdots \\ A_{s-1} & A_{s-1}v_{s-1} & B_{s-1} & B_{s-1}v_{s-1} & C_{s-1} & C_{s-1}v_{s-1} & \sqrt{A_{s-1}^{2}+B_{s-1}^{2}+C_{s-1}^{2}}\\ A_{s} & A_{s}v_{s} & B_{s} & B_{s}v_{s} & C_{s} & C_{s}v_{s} & \sqrt{A_{s}^{2}+B_{s}^{2}+C_{s}^{2}} \end{array}\right]$$

We have
(5)$$\left[\begin{array}{c} x_{0}\\ l\\ y_{0}\\ m\\ z_{0}\\ n\\ d \end{array}\right]={\left(Q^{T}Q\right)}^{-1}Q^{T}{\left[\begin{array}{c} -D_{1}\\ -D_{2}\\ \cdots \\ -D_{r}\\ \cdots \\ -D_{s-1}\\ -D_{s} \end{array}\right]}^{T}$$

Since each of the equations comes from a separate random attitude of the plane, they are arithmetically equivalent. After the system of equations is constructed, *d* can be solved, and then, we try to delete the *r* row of *Q* (r=1,…,s) and the corresponding $D_{r}$ in order from 1 to *s*, and re-solve *d*. If the absolute difference between *d* and its measured value decreases, the deletion is retained; otherwise, the operation is rolled back and redone in the next (*r* + 1) row.

When the number of deletions reaches a certain threshold *M* or the difference between the solved *d* and its measured value is less than a certain threshold Thresh, the above operation is terminated and all unknown variables are solved with the remaining equations. Although we cannot know the error of ($x_{0}$, $y_{0}$, $z_{0}$, l,m,n) from the solution result, the difference between *d* and its measurement can be used for the evaluation of the accuracy of the entire solution. The whole procedure is shown in Figure 2.

The above model is different from the traditional plane calibration in two ways:The plate thickness *d* is newly introduced as an unknown variable;The solution error of the plate thickness *d* is used as the basis for equation screening, which is also the main means of our method to ensure accuracy.

## 3. Algorithm Simulation

### 3.1. Simulation Preparation

To evaluate the proposed calibration model, an algorithmic simulation was conducted, in which simulated data for calibration calculations were generated and fed into the calibration model.

For laser displacement sensors using the trigonometric principle, the maximum value α of the angle between the laser beam and the normal vector of the surface to be measured is limited [17], and is usually less than $10^{\circ }$. At the same time, in order to avoid the strangeness of the equations, there should be a minimum β limit of an angle between any two postures in the process of changing the attitude of the wafer. The above constitutes the first two constraints. Then, the LDS usually has a linearity value, indicating the difference between its indication and the ideal curve (a straight line with a slope), which can serve as an error in its results.

In order to find out the factors that affect the calibration, we modeled the calibration process, in which we had an LDS and a thick plate with two sides that were parallel. The LDS was abstracted as a unit with a coordinate origin (representing the laser beam origin or sensor origin), a measurement vector (representing the laser beam), and a measurement voltage (representing the actual voltage value and which determined the length of the measurement vector). The measured voltage was added with a random error of 0.1% to simulate a real LDS. The visualization of the simulation is shown in Figure 3. In the visualization, the laser beam, the laser beam origin, the laser projection point, the laser-projected plane, and the additionally measured plane from two random generations are shown. The laser beam origin and the laser beam remained unchanged, whereas the laser-projected plane and the additionally measured plane were different. Thus, the laser projection point was also different. The data required for calibration came from this model, and all data were subject to the necessary random errors.

In our simulation, we preset the sensor’s laser beam vector and origin coordinates, as well as the thickness of the plate. To avoid the influence of special values, a random change was attached to the preset values during each individual simulation. To generate the parameters of each plate’s pose and position, a set of $A_{s}$, $B_{s}$, and $C_{s}$ was randomly generated, and, then, the measurement of voltage $v_{s}$ was obtained. $D_{s}$ was calculated by using $A_{s}$, $B_{s}$, $C_{s}$, and $v_{s}$ with preset values. Thus far, we could obtain the ideal parameters of the LDS and plate under the current plate pose and position. Then, we applied random errors to the planar parameters and voltages, where the voltage error was set according to the physical parameter of the LDS, which was 0.1%, and the planar parameter error δ was the variable factor we were concerned with. The error level was constant in the same simulation. The generation of random $A_{s}$, $B_{s}$, $C_{s}$ and $v_{s}$ was repeated until the number of equations reached a specific value *K*, which was another concerning factor. During the generation process, $A_{s}$, $B_{s}$, and $C_{s}$ underwent screening, which was based on the maximum and minimum angles mentioned earlier. Among them, the maximum angle limited the angle between the normal vector of the plane (i.e., ($A_{s}$, $B_{s}$, and $C_{s}$)) and the preset sensor measurement vector (i.e., the laser beam vector in the real LDS), and the minimum angle limited the minimum angle value of angles between the normal vector of the newly generated plane and that of all planes that already existed. Only the $A_{s}$, $B_{s}$, and $C_{s}$ that passed the maximum and minimum angle screening could be used in the subsequent procedure. The whole procedure is shown in Figure 4. The intermediate variables *a*, b,c, and *r* were used to generate the plane parameters.

Once enough data were obtained, they were substituted into the calibration program, the solved variables of $x_{0}$, $y_{0}$, $z_{0}$, *l*, *m*, *n*, and *d* compared with the preset ones, and the ratio of the error and the preset value (i.e. the relative error) recorded. With the same set of α, β, δ, and *K*, this process was repeated 100,000 times, resulting in a normal distribution of errors. An example of $x_{0}$ is shown in Figure 5.

The purpose of the simulation experiment was as follows:Find out if, and how, variables of interest affected calibration errors;Explore the influence of the change of plate thickness *d* on the calibration errors;Compare calibration error distributions with and without equation screening via the error of *d*.

### 3.2. Simulation Results

As a simulation of a real-world scenario (which is shown in Section 4), we set the initial parameters of the sensor to ($x_{0}=15.0$ mm, $y_{0}=15.0$ mm, $z_{0}=80.0$ mm, l=0.6, m=0.6, n=1.8, d=0.6 mm). As mentioned earlier, these presets were applied with a random variation for each simulation experiment to avoid being specific to certain data. It was clear that $x_{0}$ and $y_{0}$ were arithmetically equivalent, as were *l* and *m*; thus, they were put together when the results were presented.

#### 3.2.1. Error Distribution under Different Factors of α, β, δ, and *K*

For simplicity, we only changed one factor at a time. Additionally, as a phenomenon that we observed by chance in the experiments, the value of $d+z_{0}$ had good resistance to various interfering factors, and thus, we also listed $d+z_{0}$ as an observation object. Figure 6 shows the error distribution of the calibration results under different values of maximum angle limits α, where the value of α increased from $7.0^{\circ }$ to $9.0^{\circ }$ in equal steps of $0.5^{\circ }$. In this set of experiments, $\beta =1.0^{\circ }$, δ=0.05%, and K=50. It is worth noting that with the increase in the maximum angle limit α, the calibration error tended to decrease. As the maximum angle limit increased, the coefficient difference between equations also increased, and, therefore, the probability of obtaining a unique solution also increased. In the least squares method, this meant a reduction in the solution error.

Figure 7 shows the error distribution of the calibration results under different values of minimum angle limits β, where the value of β increased from $0.1^{\circ }$ to $0.9^{\circ }$ in equal steps of $0.1^{\circ }$. In this set of experiments, $\alpha =9.0^{\circ }$, δ=0.05%, and K=50. Similar to the maximum angle limit α, the increase in the minimum angle limit β also led to a decrease in the calibration error. At the same time, under the condition that the maximum angle limit α was unchanged, the increase in the minimum angle limit β compressed the feasible attitude space, thus reducing the average difference between the various calibration postures. Therefore, the error did not decrease significantly for $z_{0}$ and *d*.

Figure 8 shows the error distribution of the calibration results under different values of plane parameter error percentages δ, where the value of δ increased from 0.05% to 0.23% in equal steps of 0.02%. In this set of experiments, $\alpha =9.0^{\circ }$, $\beta =1.0^{\circ }$, and K=50. Obviously, the more accurate the plane parameter obtained, the smaller the calibration error. Therefore, for better calibration results, flatter planes and higher precision planar measurements are needed.

Figure 9 shows the error distribution of the calibration results under different values of equation volumes *K*, where the value of *K* increased from 35 to 125 in equal steps of 15. In this set of experiments, $\alpha =9.0^{\circ }$, $\beta =1.0^{\circ }$, and δ=0.05%. As the number of equations increased, the calibration error decreased, but for *z* and *d*, we saw a phenomenon similar to when the minimum angle limit increased. The reason was similar to that of Figure 6: the maximum and minimum angle limits did not change, and the increase in the amount of data led to a decrease in the average difference between the calibration attitudes.

#### 3.2.2. Error Distribution under Different *d*

Figure 10 shows the error distribution of the calibration results under different values of plate thicknesses *d*, where the value of *d* increased from 0.2 mm to 1.8 mm in equal steps of 0.2 mm. In this set of experiments, K=50, $\alpha =9.0^{\circ }$, $\beta =1.0^{\circ }$, and δ=0.05%. As a newly introduced parameter, the size of *d* did not have a significant effect on the solution accuracy, except for its own accuracy. Since the error was normalized, it could be inferred that the error value of *d* did not change with the change in *d*.

#### 3.2.3. Error Distribution with and without Equation Screening

Figure 11 shows the error distribution of the calibration results under different equation volumes of *K* without (tagged with origin) and with (tagged with ours) equation screening, where the value of *K* increased from 35 to 125 in equal steps of 15. In this set of experiments, $\alpha =9.0^{\circ }$, $\beta =1.0^{\circ }$, and δ=0.05%.

It is clear that the solution accuracy of $x_{0}$, $y_{0}$, $z_{0}$, *d*, and $d+z_{0}$ was significantly improved by using equation screening. There was a slight drop in the solution accuracy for *l*, *m*, and *n*, which came from a decline in the number of valid equations. In the case of a large enough number of equations, the difference brought by screening was small enough to be negligible.

Overall, it could be concluded that our screening method was very effective, and could significantly improve the accuracy of the LDS’s position and pose calibration, even when using only the linear least squares method. Therefore, this method was implemented in a real system. According to the simulation, we needed a calibration plate that was flat enough and parallel on both sides. To achieve this, we investigated a variety of possible objects, including metal machined parts, float glass, and wafers. The flatness and parallelism of machined parts were difficult to guarantee, and transparent float glass was difficult to measure. On the other hand, wafers had a sufficiently high flatness and parallelism, and, at the same time, were easy to obtainm due to being a raw material of the semiconductor industry.

The general conclusion we could draw from the simulation was that, in order to obtain better calibration results, we should increase α, as much as possible (without exceeding the LDS’s technical requirements), and reduce δ as much as possible (if it can be achieved), whereas the optimal value of β was $0.6^{\circ }$, and *K* should be greater than, or equal to, 100 to have significant advantages over the original method in every parameter.

After obtaining the law of the influence of different parameters on the calibration error, we explored the measurement accuracy of the surface normal at different LDS calibration error levels through a simple calculation program. Simply put, the parameters with errors were substituted into the normal calculation equation and compared with the calculated result without errors.

After a series of attempts, a conclusion was drawn. To achieve a surface normal measurement accuracy of $0.1^{\circ }$, the parameters required for calibration were as follows: δ≤0.05%, K≥100, $\beta \geq 0.6^{\circ }$, and $\alpha \leq 9.0^{\circ }$. The requirement for α was to consider the situation of multiple sensors, in which there were angles between the sensors, and each sensor had to maintain an angle of $10^{\circ }$, or less, with regard to the plane normal. Thus, there should be a safe gap between α and the technical requirements of LDS on the maximum angle (which was usually $10^{\circ }$).

## 4. Experiment

The object of the actual calibration experiment was a hand-held instrument aimed at measuring the verticality of the hole, as shown in Figure 12. At the end of the instrument were four LDSs distributed equidistantly around the circumference, and on the axis of the circumference was a rod with three protruding claws to accommodate holes of different diameters. When measuring, the rod probed into the hole and the three claws extended out; thus, the rod and the hole were coaxial. At this time, the four LDSs measured the distances from themselves to the surface around the hole and calculated the 3D coordinates of the four measured points, according to the origin coordinates and laser beam direction vector obtained by calibration. Then, the surface normal was obtained by plane fitting, and its angle with the axial direction of the rod was calculated. The axial direction of the rod was measured prior and recorded with the instrument software. The four LDSs were the Panasonic HG-C1030 model, where the measurement center was 30 mm away from each sensor’s origin. The measurement range of the sensors was 30 ± 5 mm, and the distance value from −5 mm to 5 mm was represented by a 0–5 V analog voltage with a nonlinearity of ±0.1% in the whole range. The circuit board in the handle read, digitized, and output the voltage values of the four sensors. The goal of calibration was to obtain the origin coordinates and laser beam direction vectors of the four LDSs. As can be seen, this instrument worked independently and was not attached to a coordinate system. Therefore, its calibration had to be exogenous. This was why we developed the proposed calibration method.

The flat plate used for calibration was a 650 μm thick double-side-grinded monocrystalline silicon wafer, with a global flatness, thickness non-uniformity, and surface roughness less than 3 μm. The wafer was chosen for its easy access and high planarization performance, which met the requirements of planar parameter errors. In the calibration method being performed from both sides, a 6-inch wafer was sufficient to meet the requirement of the observable area.

In order to achieve better reproducibility of the attitude transformation, the wafer was installed on an electric motor-driven platform, which had six motors and controlled the attitude and position of the wafer within a certain angle range, as shown in Figure 13. The platform consisted of a fixed frame and a mobile frame. The three motors mounted on the fixed frame drove the mobile frame linearly along three rails to achieve the change in the planar equation parameter *D*. The three motors mounted on the mobile frame drove the wafer holder to change the pose to achieve the change in the planar equation parameters *A, B,* and *C*. Thus, the wafer could be directed to any pose and position within the range of the connecting rods. Motor rotation angles under each pose and position could be recorded and replayed. At the same time, they could also be randomly generated by the controlling software. The motor-driven platform was specially designed and developed for calibration by the authors. The step motors were manufactured by Makerbase and the frame consisted of aluminum profiles and 3D-printed parts. The controlling software, which had the ability of moving control, pose prediction, and data calculation, was developed under UOS (Uniontech Operating System). The platform ensured the experiments were reproducible and improved the efficiency of random plane generation, compared to manual operation. To avoid the force deformation of the wafer, the wafer holder flexibly clamped the wafer via two rubber rings and maintained a very low clamping force that only kept the wafer relatively stationary while moving. The effectiveness of this clamping method in avoiding deformation was verified by the RANSAC (random sample consensus) plane fitting of the measured point cloud of the wafer when it was fixed.

Specifically designed for calibration, the front and rear of the silicon wafer on this platform were fully open to be observed by the SLP and the LDSs at the same time. As the source of precise plane parameters, the SLP adopted the principle of a three-frequency, four-step phase shift, had two cameras with 5 million pixels, and was calibrated by a checkerboard with a pattern accuracy of ±1 μm. The SLP had a scanning volume of around 40 mm ∗ 30 mm ∗ 20 mm and achieved a measurement accuracy of 0.02 mm under the VDI/VDE 2634 standard [18]. The planar equation parameter uncertainty was less than 0.02% for multiple scans and fittings of the same surface when the wafer was fixed. The SLP is shown in Figure 14.

The complete experiment setup is shown in Figure 15. In simple terms, this seemingly complicated scene only contained three elements:The sensors to be calibrated;The wafer, having a posture that could be changed;The SLP.

The motor platform could be replaced by a simple stand, and the SLP could be replaced by any device capable of measuring planar parameters, such as three or four calibrated LDSs.

In the process of changing the position and pose of the wafer, it might exceed the range of some LDSs and fall within the range of other LDSs. All LDSs within the range could adopt this attitude to achieve the multiplexing of calibration data among the sensors. Thus, in actual experiments, the ratio of this multiplexing could be as high as 80% to 90%. The uncertainty of the parameter measured on the same plane by the SLP was less than 0.05%, reaching the optimal state of the simulation. The data volume was set to 100. A simplified diagram of the calibration procedure is shown in Figure 16.

As a verification of the calibration results, the calibrated LDSs and the SLP were used to measure the wafer’s surface normal vector as a basis for the initial posture (when all motor rotation angles were $0^{\circ }$), and then, they were used to measure the wafer under each of the 115 different poses, simultaneously. The angle between the normal vector of each pose and the basis was output separately, the result from the SLP was used as a reference, and the result from the LDSs was used as the measurement value. A simplified diagram of the validation procedure is shown in Figure 17.

The absolute differences between the measurement values and the corresponding references were regarded as the calibration error, the distribution of which is shown in Figure 18. The results conformed to a normal distribution. As can be seen from the figure, most of the results were distributed in the $0^{\circ }$ to $0.08^{\circ }$ range, and, out of 115 validations, only 3 were distributed in the $0.08^{\circ }$ to $0.10^{\circ }$ range, whereas only one reached $0.10^{\circ }$.

## 5. Conclusions

This paper presented a novel method to calibrate laser displacement sensors by observing the opposite sides of a wafer and reducing the calibration error by optimizing the results of the wafer thickness. A series of simulations was conducted to explore the effect of several factors on the calibration error and to validate the superiority of the proposed method, compared to the traditional one. The calibration method was applied to a handheld instrument with four LDSs. The surface normal measurement results of the instrument, after calibration, were compared with those of a SLP in 115 validations, and the max error was $0.1^{\circ }$. The proposed method is valid for laser displacement sensors and also instructive for the calibration of other similar sensors. Since the method presented in this paper only involves equation screening in linear solving, the screened equations can still be constructed in a nonlinear form to further improve accuracy.

## Figures and Tables

**Figure 1 sensors-23-01762-f001:**
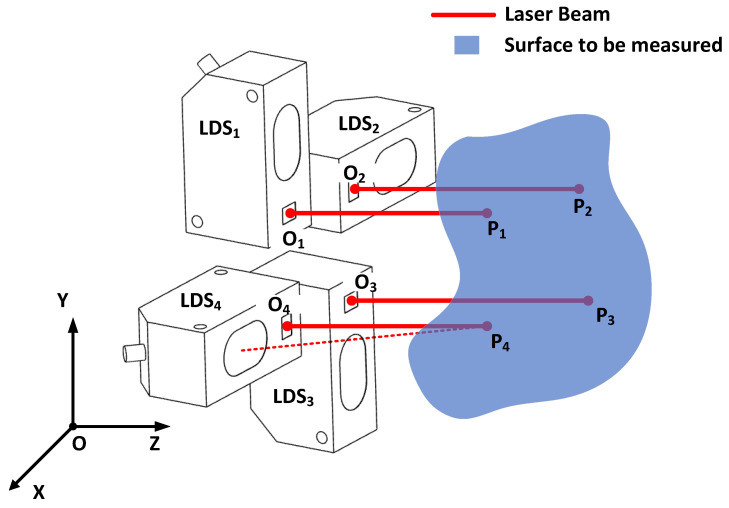
A sketch of the surface normal measurement scene.

**Figure 2 sensors-23-01762-f002:**
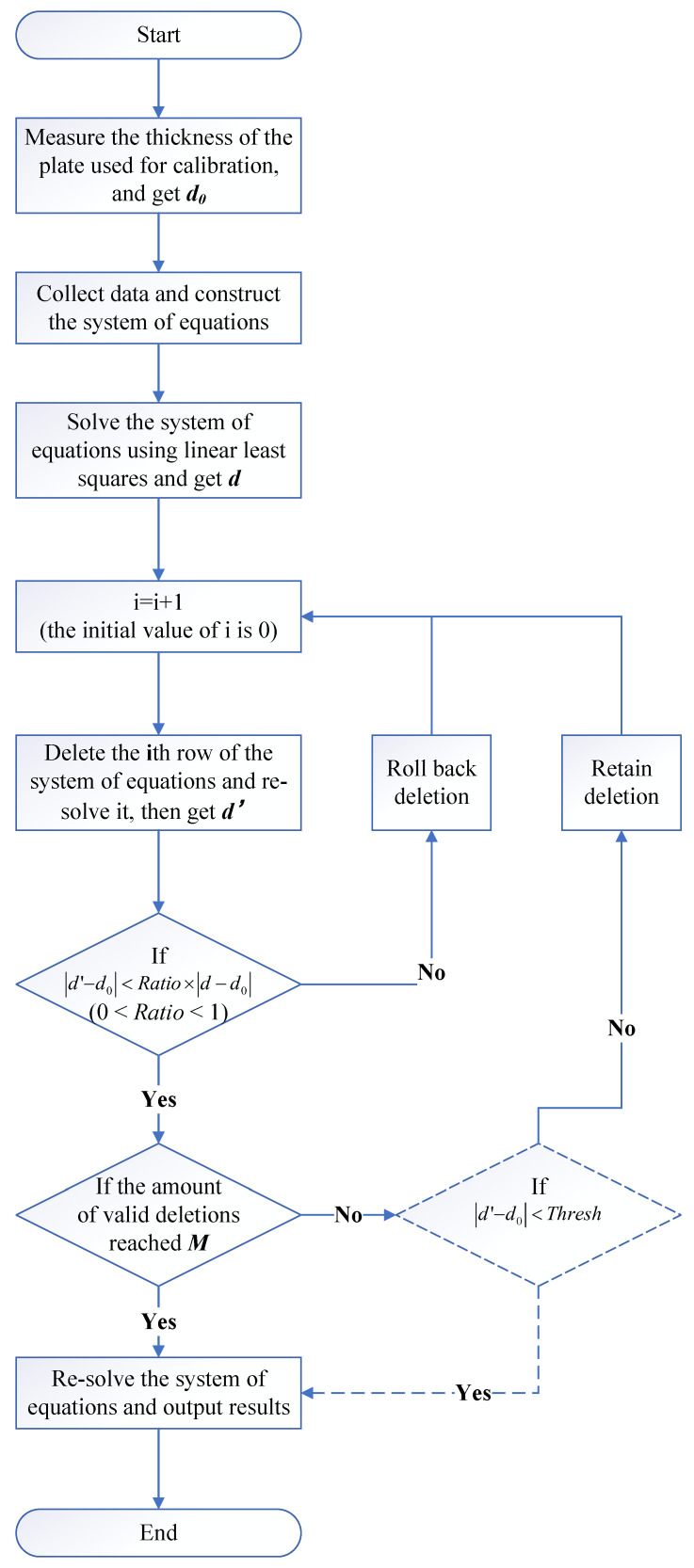
Whole procedure of the proposed calibration algorithm.

**Figure 3 sensors-23-01762-f003:**
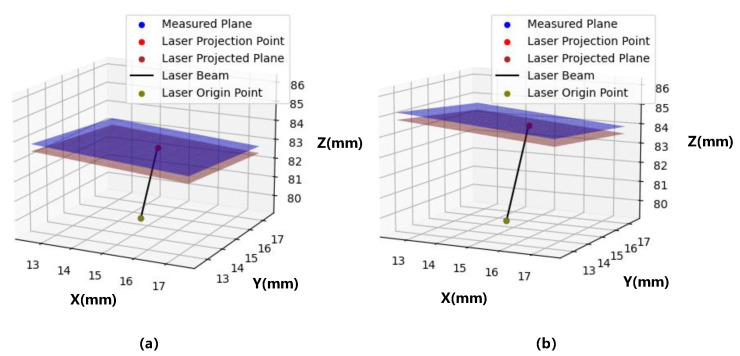
Visualization of the random generations (**a**,**b**) in a Cartesian coordinate system (axes are measured in millimeters).

**Figure 4 sensors-23-01762-f004:**
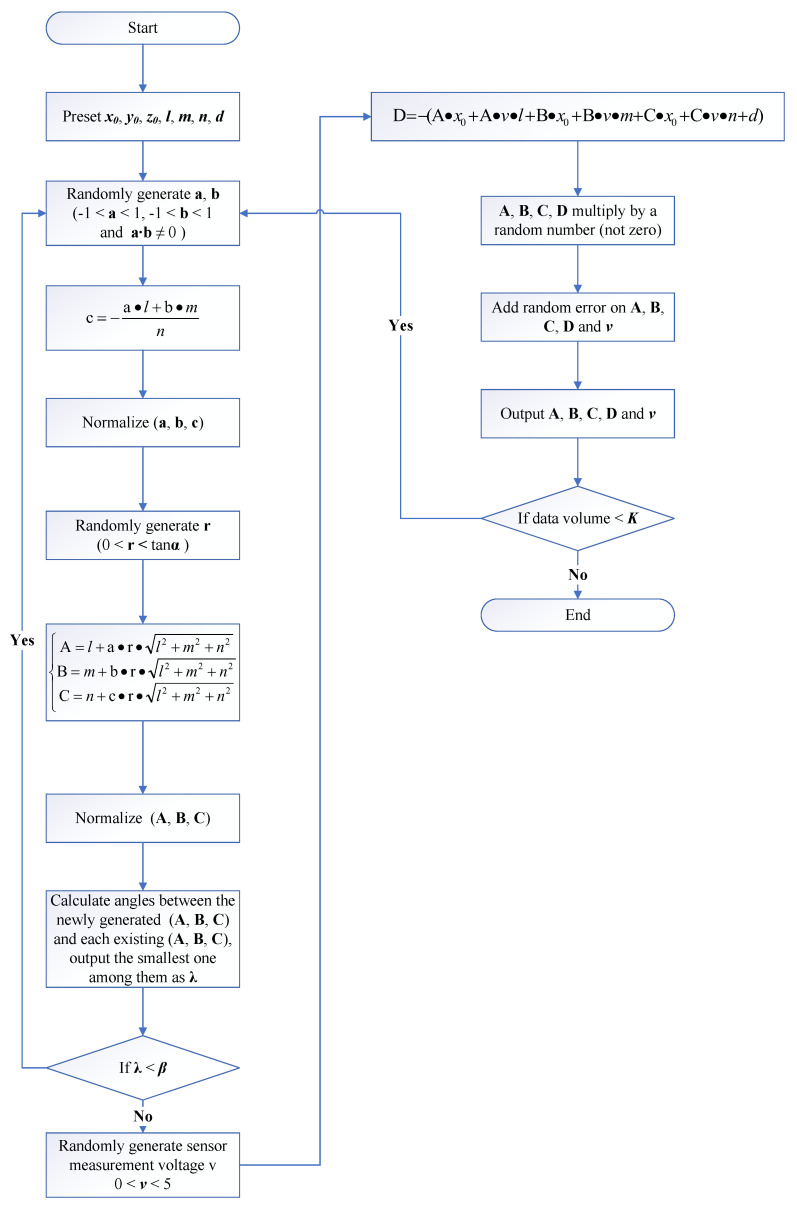
Whole procedure of the simulation algorithm.

**Figure 5 sensors-23-01762-f005:**
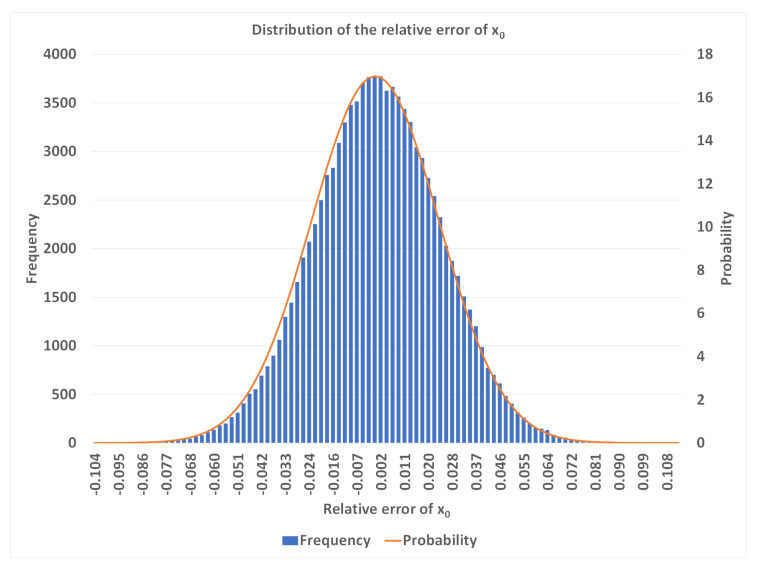
Normal distribution of the relative error of $x_{0}$, with $\alpha =9.0^{\circ }$, $\beta =1.0^{\circ }$, δ=0.05%, and K=50.

**Figure 6 sensors-23-01762-f006:**
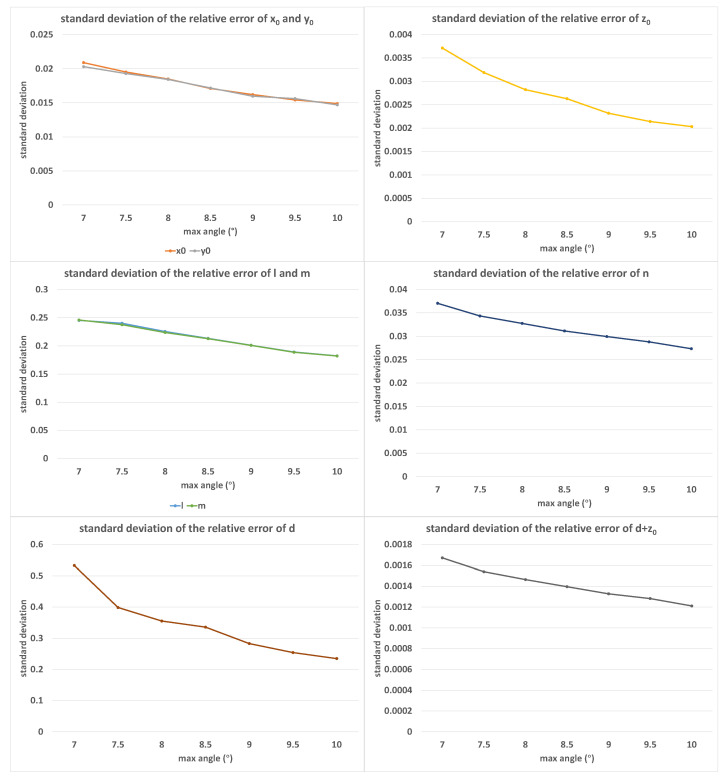
The standard deviations of relative errors of ($x_{0}$, $y_{0}$, $z_{0}$, *l*, *m*, *n*, *d*, $d+z_{0}$) with the increase in α; the horizontal axis was measured in degrees.

**Figure 7 sensors-23-01762-f007:**
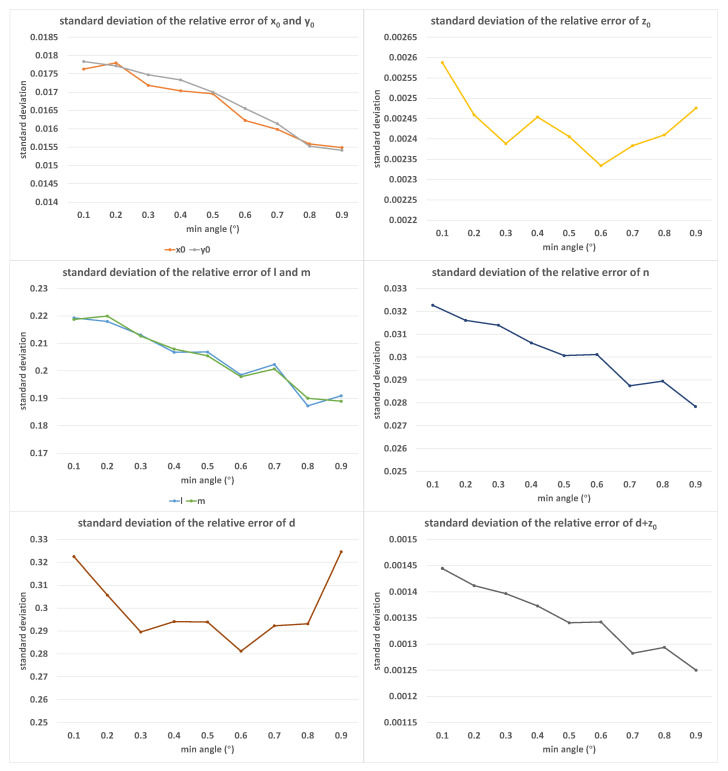
The standard deviations of relative errors of ($x_{0}$, $y_{0}$, $z_{0}$, *l*, *m*, *n*, *d*, $d+z_{0}$) with the increase in β; the horizontal axis was measured in degrees.

**Figure 8 sensors-23-01762-f008:**
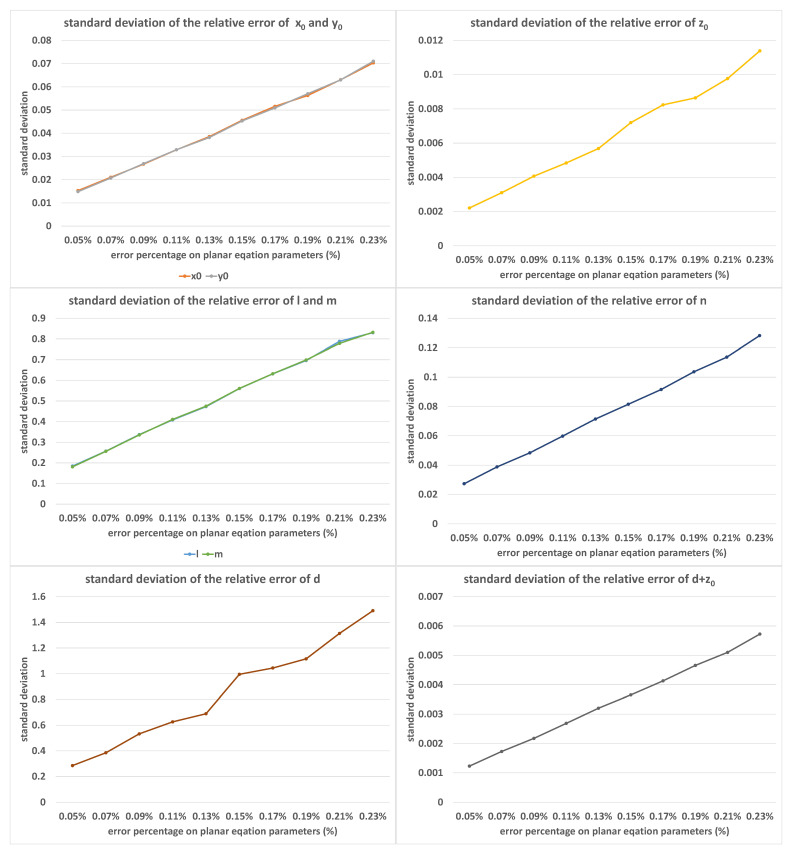
The standard deviations of relative errors of ($x_{0}$, $y_{0}$, $z_{0}$, *l*, *m*, *n*, *d*, $d+z_{0}$) with the increase in δ; the horizontal axis was measured as percentages.

**Figure 9 sensors-23-01762-f009:**
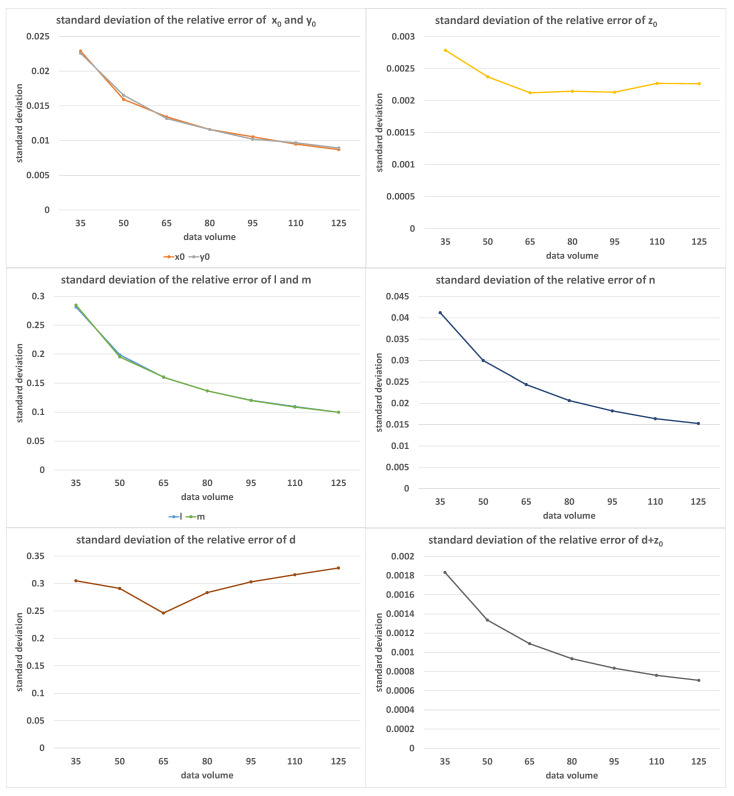
The standard deviations of relative errors of ($x_{0}$, $y_{0}$, $z_{0}$, *l*, *m*, *n*, *d*, $d+z_{0}$) with the increase in *K*; the horizontal axis was measured in equation volume.

**Figure 10 sensors-23-01762-f010:**
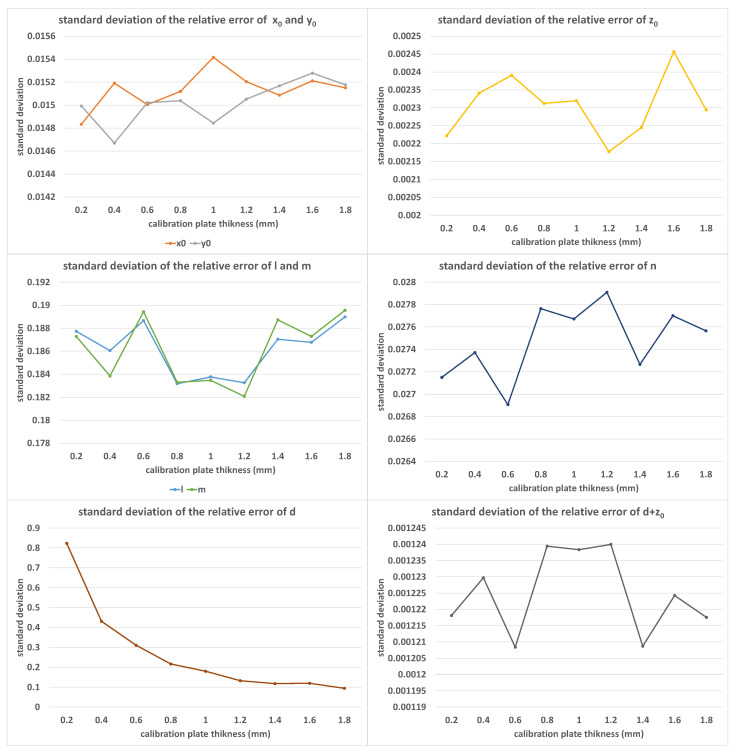
The standard deviations of relative errors of ($x_{0}$, $y_{0}$, $z_{0}$, *l*, *m*, *n*, *d*, $d+z_{0}$) with the increase in *d*; the horizontal axis is measured in millimeters.

**Figure 11 sensors-23-01762-f011:**
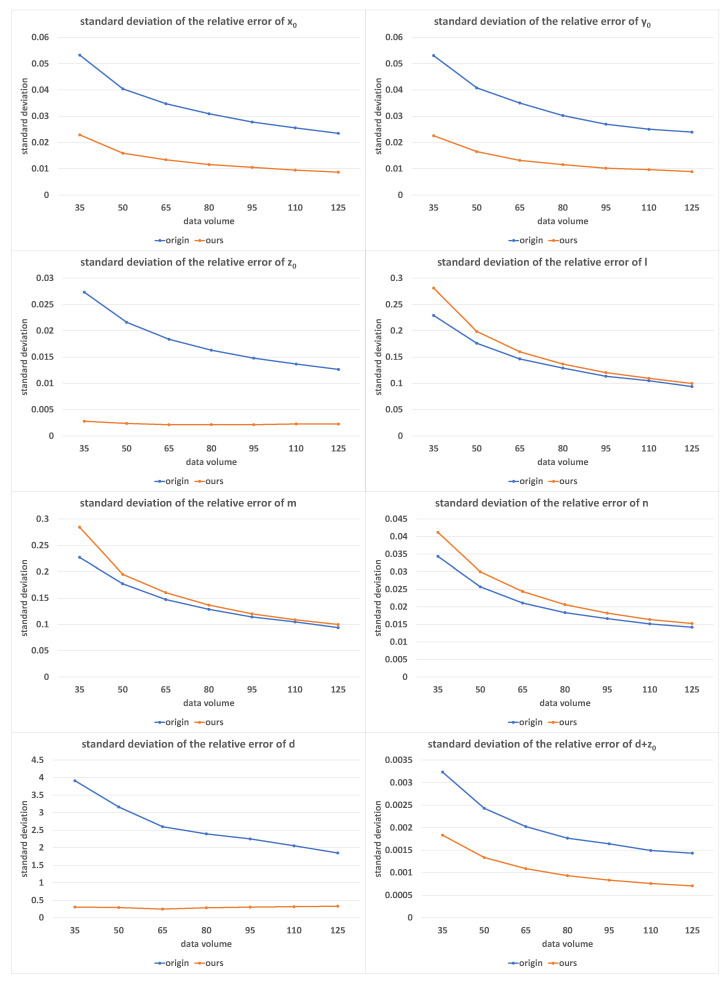
The standard deviations of relative errors of ($x_{0}$, $y_{0}$, $z_{0}$, *l*, *m*, *n*, *d*, $d+z_{0}$) with the increase in *K*; with and without equation screening; the horizontal axis was measured in equation volume.

**Figure 12 sensors-23-01762-f012:**
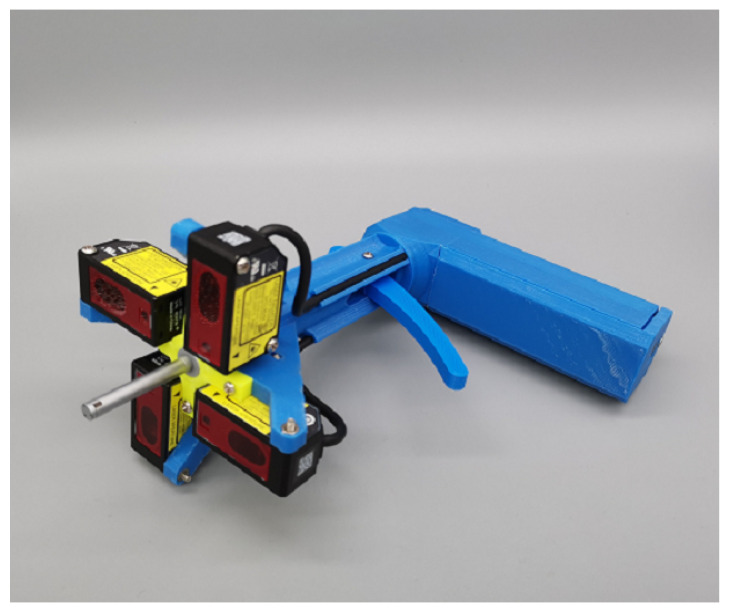
The calibration object: a handheld hole–verticality measurement instrument.

**Figure 13 sensors-23-01762-f013:**
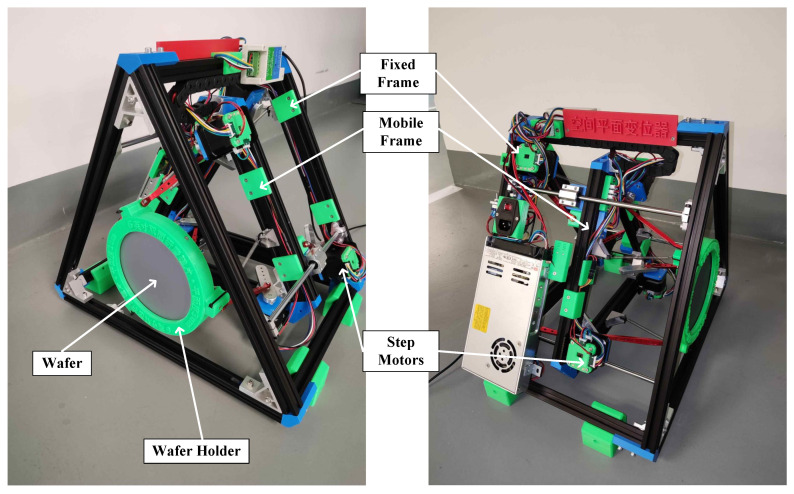
The electric motor-driven platform for plate attitude transformation.

**Figure 14 sensors-23-01762-f014:**
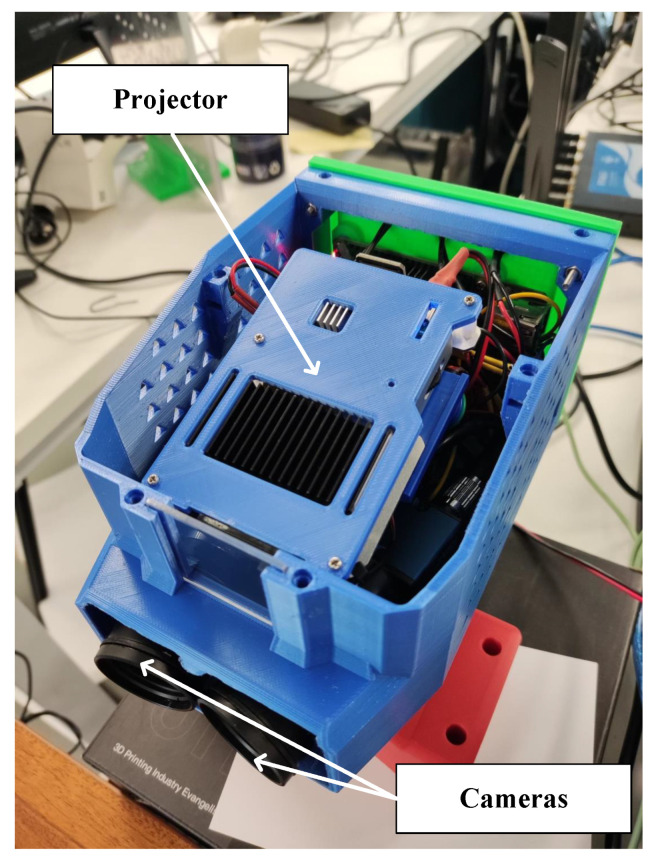
The structured light probe.

**Figure 15 sensors-23-01762-f015:**
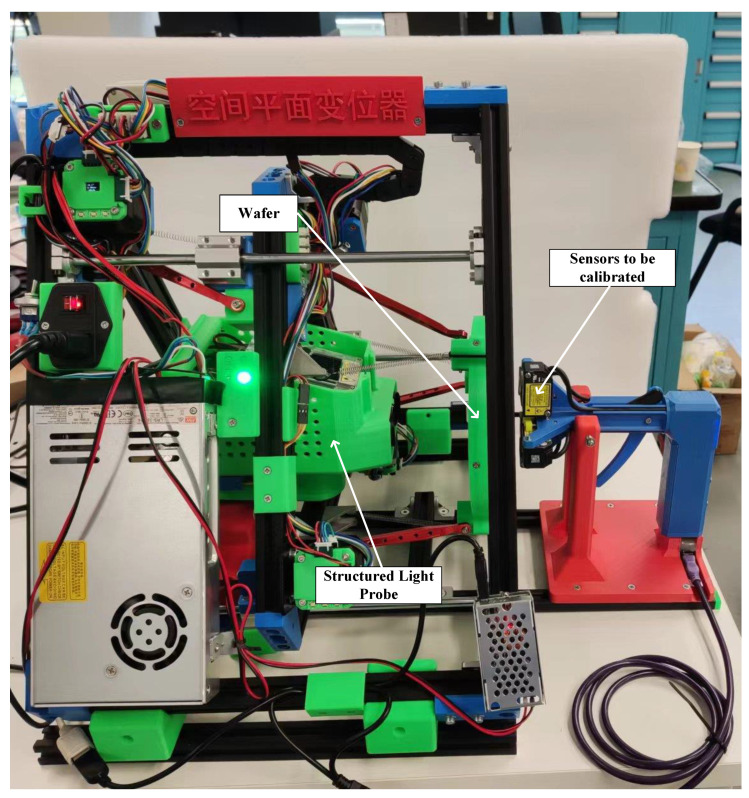
The experimental setup (characters in the nameplate means “Spacial Plane Position and Attitude Adjustment Device”).

**Figure 16 sensors-23-01762-f016:**
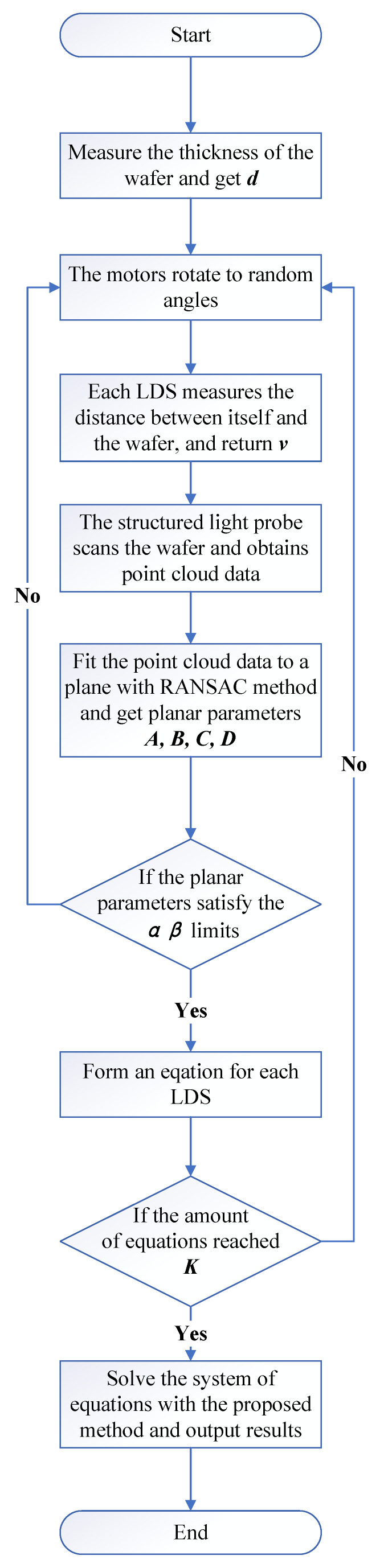
The calibration procedure of the experiment.

**Figure 17 sensors-23-01762-f017:**
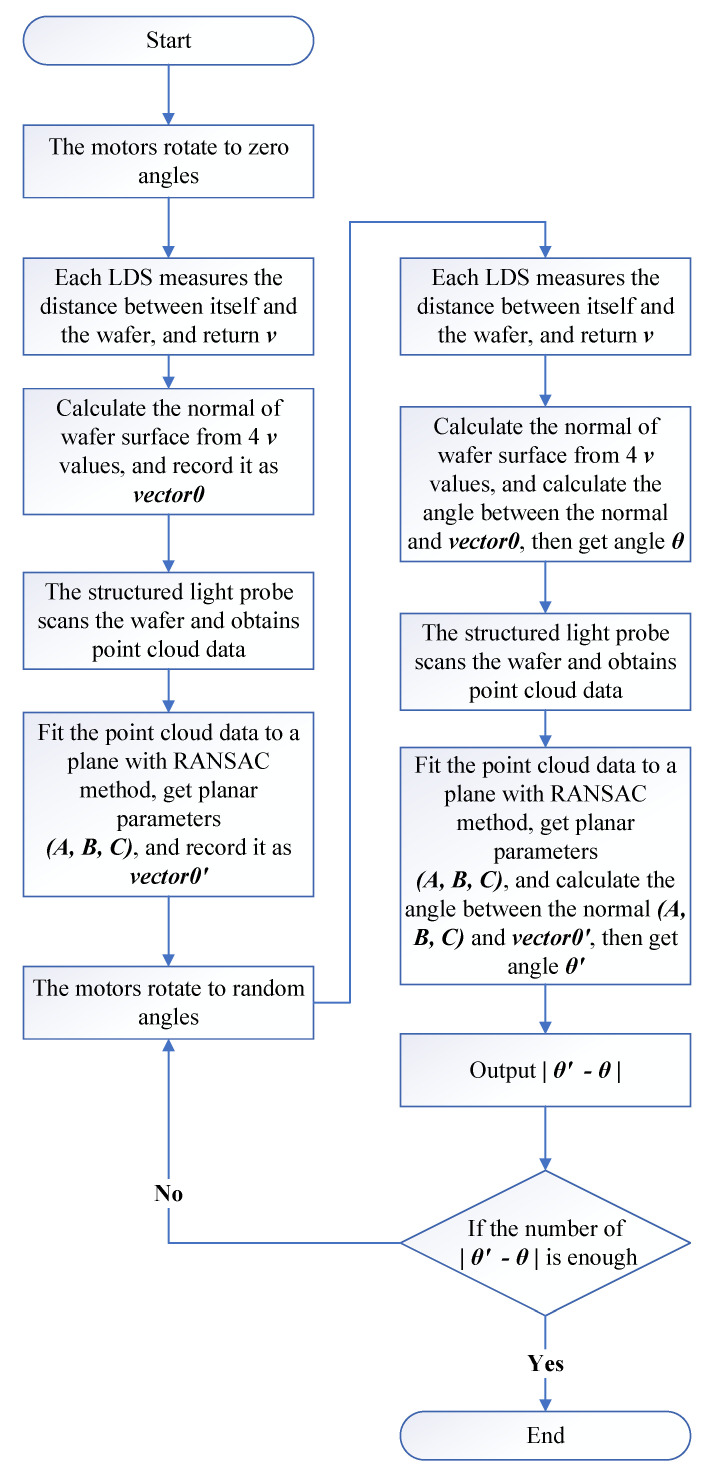
The validation procedure of the experiment.

**Figure 18 sensors-23-01762-f018:**
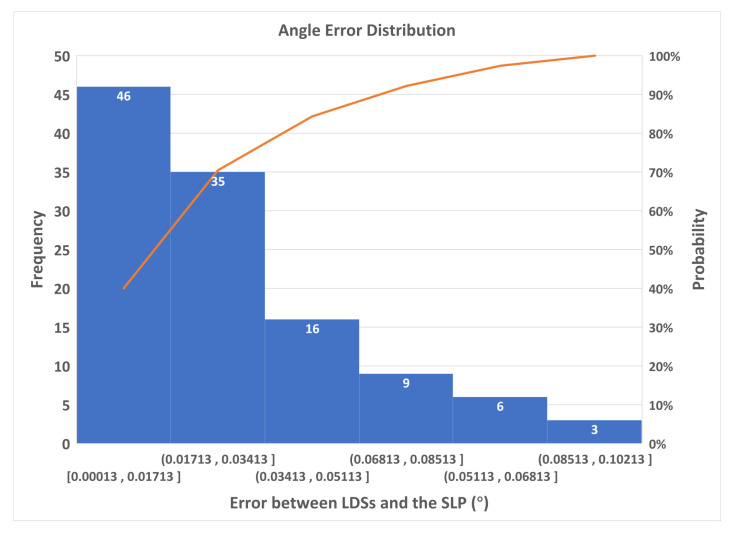
The distribution of the calibration errors.

## Data Availability

The data that support the findings of this study are available from the corresponding author, upon reasonable request.

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
