# Peer review of "A Novel Method for Pose and Position Calibration of Laser Displacement Sensors"

_sensors, 2023, doi:10.3390/s23041762_

Round 1
Reviewer 1 Report
Dear authors,
thank you for the manuscript "A Novel Method for Pose and Position Calibration of Laser Displacement Sensors".
I have been reviewing the document and - up to what could be figured out by now - regard the significance of the content as high.
However, it seems that
1. the introduction may cover a extensive review of current developments incl. references
2. the manuscript needs to be revised to be more clearly. A supporting sketch for defining the parameters is recommended
3. the manuscript should be re-checked in terms of spell and grammar checking (e.g. doubled "the the" on line 47) also in Figures (fig 1)
I recommend to include these issues in future revisions. Thank you
Reviewer 2 Report
In this manuscript, the authors try to introduces a new parameter to the traditional calibration procedure to reduce the calibration error and explores the factors affecting calibration using Monte Carlo method. The author verified through simulation that the solution accuracy of x0, y0, z0, d and d+z has been significantly improved by using this method. Moreover, the method has been applied to a practical system and the error results have been obtained.
Overall the article is clear and the work is presented transparently. However, before publishing, I hope the author can carefully revise their manuscript considering the following comment:
1. It is mentioned in the manuscript that the thickness of the double-side grinded monocrys-talline silicon wafer is obtained by scanning the probe. However, the basic data parameters of the experiment, such as the resolution and accuracy of the probe, have not been shown.
2. It has been mentioned that thickness non-uniformity and surface roughness of the wafer, however, the profile error and clamping error of the wafer has been ignored. This should be discussed.
3. When the wafer is changing its attitude, the laser beam is always in the normal direction of the chip? Please explain whether the measurement has an angle range.
4. Please supplement the schematic diagram of the measurement process to explain the working principle of the device measurement.
5. Fig. 5-10, the information of coordinate axis is not marked. Adjust the Fig. 4 and use the original figure.
Reviewer 3 Report
The authors describe a novel method to calibrate a laser displacement sensor to determine location and pose using a planar surface. The paper describes the need, outlines the algorithm, performs a simulation using the algorithm to do a a sensitivity analysis of the various calculation parameters, and finally does an experimental demonstration of the approach. All of the components are there for a good paper but there are some issues. Such as:
Fig. 2 needs more explanation and an improved caption. The text says a visualtization of the simulation is shown in Fig. 2. All I see is two planes in the left and right plots (which should be labeled a and b or something similar) with no explantion of what is being conveyed. All I see are two planes with different orientations.
Fig. 3: Please provide an explanation for the role of the lower case symbols a, b and c as there is no explanation in the text, just in the block diagram. It seems they are used to generate and perturb and A, B, and C coefficients to achieve pseudo random inputs for the simulation.
Line 147: Are not l, m, and n direction cosines? How can n be 1.8, i.e greater than one?
Line 249: How can the wafer have 115 poses simultaneously if the motors must move the wafer for each pose?
Finally, Fig. 15 which should be the highlight of your paper is very confusing to interpret. More explantion is needed in the caption to highlight your results. How does the reader interpret the error between the algorithm calculation and ground truth? Please add more detail to your explanation, in the conclusion you state your maximum error is 0.1 degrees. How do I get that from Fig. 15?
Reviewer 4 Report
The authors present a study to overcome current limitations of existing calibration procedures for laser distance sensors by proposing a new calibration schema. However, when it comes to the details, the paper is missing clarity, correctness, completeness and readability.
For instance: The addressed current drawbacks are somewhat fuzzy. What means '... is more sensitive to errors...'? In comparison with what? What 'laser beam vector' of what sensor type/sensor?
As a result of the poor and not comprehensible motivation, the aim of the paper remains a riddle.
Then, there is no sketch about the proposed principle, which makes it difficult to follow the explanations in section 2. Again, what laser beam of what sensor in what orientation? What is the aimed quantity of the calibration? Indeed, the complete paper is currently structured and written more as a report rather than a scientific paper. Please make sure, that your aim of the derivation that follows is clear in the beginning, not at the end. Currently I doubt the novelty is worth to be published in a scientific journal.
In what way is the simulation fitting to the later experimental results? Unclear... What about the units of the parameter ranges (Fig. 2)? The diagrams quality in section 3 is low. Axis labels are not given, numbers are very small. No units are given in the diagrams, only in the figure captions. Why are the chosen parameter ranges of interest? What are the universal findings? In what way are the simulations related to the experimental results in section 4?
In section 4, the photos show no scale. The photos are too many since the respective imformation transderred to the reader is low. Only a single points seems to be validated and the results of section 3 remain un-validated.
As a result, the manuscript quality is too low and therefore I must recommend rejection.
Round 2
Reviewer 1 Report
Dear Authors.
Thank you for the revision of your manuscript "A Novel Method for Pose and Position Calibration of Laser Displacement Sensors". In this work you describe an improved method for laser displacement sensing.
The paper is clearly structured and easy to understand. Thank you.
However, I suggest to check my list of comments and address them in a minor revision:
Page2
Line 52: expressions "It works well" are a qualitative measure. Please specify clearly
Line 56: expressions "this work is significant, but the scenario is limited". Again: qualitative measure. Please specify clearly
Line 85/6: Please specify: which kind of error did you achieve? Please give a reference
Page 3
Line 94: "enough" what is the criterion for enough equations? Giv a reference to later explanations
Line 106: derive instead of achieve
Line 122: change proposed: This is the basic equation for calibration.
Line 122: define D please
Line 136/7: replace "The minimum value of s is 7" by The minimum value of s is 7 to solve equation sytem 3
Page 5
Fig 1: "Re-sovle the system of equations and output results" Typo: "Re-solve"
Page 6
Line 173 remove "algorithmically"
Line 178: remove "given"
Page17
Line 323: please specify motor driven platform (type, manufacturer...)
Page18
"RANSAC" add "(random sample consensus)"
Page 21
"Sovle the system of equations with the proposed method and output results" Type: "Solve"
The scientific soundness of some parts of the paper could and should be improved:
1. In several paragraphs of Sec. 3.2 you the authors state "We believe.....". Science is not religion. Please give evidence! (in all cases)
e.g. change to "Indicators for...", "Evidence is given by..."
2. On page 17 the authors praise their method by "Thanks to our...". Please be more neutral. eg. by "In calibration method being performed from both sides"
Reviewer 3 Report
I appreciate the authors' willingness to consider my comments and address them in a satisfactory manner. I suggest that the manuscript be published in its present improved form.
Author Response
Thank you very much for your affirmation, and we have checked the grammar and spelling errors again in this revision.
Reviewer 4 Report
While the authors made a good attempt to improve the paper quality, and it has improved indeed, the fuzzyiness still remains. How is surface-normal measurement related to laser displacement sensors? I understand the LDS are used to calibrate the sensor's (which?) position, but it is not clearly described. The aims are with pre-references to sections and so detailed and still not clear, that the crucial basis for a scientific article is missing. No sketch of the measurement principle has been included as suggested and the approach remains mysterious. While I have no doubt that the auther idea is good and they have everything in mind, it is just not readable in the article. Therefore I again must recommend to reject the manuscript.
Round 3
Reviewer 4 Report
The authors made an excellent attempt to address my concern and the paper has substantially been improved. Now I can recommend publication.